# Unraveling Infant Feeding Practices Among Syrian Refugees in Türkiye: A Cross-Sectional Study

**DOI:** 10.3390/nu17040684

**Published:** 2025-02-14

**Authors:** Meryem Erat Nergiz, Sıddika Songül Yalçın, Suzan Yalçın

**Affiliations:** 1Department of Social Pediatrics, Institute of Child Health, Hacettepe University, Ankara 06100, Türkiye; 2Department of Pediatrics, Yenimahalle Research Hospital, Yıldırım Beyazıt University, Ankara 06010, Türkiye; 3Department of Food Hygiene and Technology, Faculty of Veterinary Medicine, Selçuk University, Konya 42003, Türkiye; syalcin@selcuk.edu.tr

**Keywords:** Syrian refugees, breastfeeding, dietary diversity, food insecurity

## Abstract

Background/Objectives: Refugee children are vulnerable in many respects. Determining their nutritional characteristics can guide the interventions that need to be developed. We aimed to determine the infant feeding characteristics of Syrian refugee mothers. Methods: Syrian health care workers administered a survey file to Syrian mothers (*n* = 210) having a child under two years of age in 39 refugee health centers from five provinces. Analysis for association was conducted using the Chi-square test and logistic regression. Results: The mean (±SD) age was 27.1 ± 6.3 years for mothers and 11.0 ± 6.2 months for infants. Thirty percent of the children had prelacteal feeding, most frequently sugary water, formula, and anise. Of all, 69% continued to be breastfed; 86% in 0–5-month-old babies and 36% in 16–23-month-old infants. Of 16–23-month-old children, 40% had not eaten any fruit and vegetables rich in vitamin A before. Multivariate analysis revealed that prelacteal feeding was associated with bottle feeding and not having lactation counseling support in the prenatal and postnatal periods (AOR: 2.61, 95% CI: 1.35–5.10; AOR: 2.79, 95% CI: 1.33–5.84). Being still-breastfed was associated with younger age (0–5 months old) and not using feeding bottles (AOR: 3.85, 95% CI: 1.47–10.10; AOR: 2.66, 95% CI: 1.35–5.21, respectively). Conclusion: Prelacteal feeding, sugary water consumption, bottle feeding, and limited dietary diversity were identified as significant nutritional issues among Syrian infants. In addition to lactation counseling, there is a need for culturally appropriate education on complementary feeding and healthy child nutrition for pregnant mothers and those with children under two years of age.

## 1. Introduction

A healthy diet is crucial for a child’s healthy growth and development. Both the World Health Organization (WHO) and the United Nations International Children’s Emergency Fund (UNICEF) advocate for early initiation of breastfeeding within one hour of birth, exclusive breastfeeding for the initial six months of life, and the introduction of nutritionally adequate and safe complementary (solid) foods at six months, accompanied by continued breastfeeding up to two years of age or beyond. WHO categorizes foods into eight groups: (1) breast milk; (2) grains, roots, and tubers; (3) legumes and nuts; (4) dairy products; (5) flesh foods; (6) eggs; (7) vitamin A-rich fruits and vegetables; and (8) other fruits and vegetables. Children aged 6–23 months should consume at least five of these groups daily for minimum dietary diversity (MDD) [1]. Although there are some studies investigating the breastfeeding and nutritional status of Syrian refugee children [2,3,4,5], there are limited studies investigating food variety in infants [3,5,6,7]. The assessment of dietary diversity relies on the 24 h recall method [1], which may not accurately reflect past feeding experience. Some children may not have tasted certain foods or they may have been fed only one or two food groups, resulting in monotonous diets [8]. The last 24 h recall period may also be skewed if the child was recently sick or facing dietary restrictions due to food allergies or intolerances. Evaluating all complementary foods introduced to the baby also indicates the mother’s competence in complementary feeding.

Given the vulnerable status of refugee populations, promoting breastfeeding and ensuring access to nutritious diets is essential for this group [9]. Türkiye is the main host country for Syrian refugees. As of 2022, there are almost 3.5 million Syrian refugees under temporary protection in Türkiye [10]. The feeding practices among children aged above six months are influenced by a conceptual framework encompassing individual, group, and societal factors [11]. Furthermore, the mothers’ beliefs and practices regarding what and how to feed their infants may be associated with the cultural and religious beliefs of society passed down through generations [3,5]. Additionally, factors such as migration, economic power, and social constraints are hypothesized to impact both the accessibility of diverse food options and the adherence to traditional feeding practices by mothers [3,5,7,12]. There are some studies investigating the breastfeeding and infant feeding characteristics of Syrian refugee children in Türkiye [2,3,4,5]. Additionally, food insecurity is a prevalent challenge among refugee populations, yet studies investigating the food security status of refugees in Türkiye remain limited [13,14]. Despite these efforts, there is a significant gap in the literature, as no comprehensive study has holistically examined breastfeeding practices, prelacteal feeding (PLF), and complementary food diversity among Syrian refugee children. Addressing this gap is essential for developing targeted strategies to improve infant nutrition and maternal feeding practices within this vulnerable population.

This study aimed to answer the following research questions.

∗What are the bottle feeding practices among Syrian refugee mothers with children under the age of two?∗What is the extent of food insecurity in infant feeding practices experienced by Syrian refugee families?∗What is the level of food diversity provided by Syrian mothers to their children under the age of two?∗What is the prevalence of lactation counseling received by Syrian refugee mothers?∗What factors are associated with PLF among Syrian refugee children?∗What factors are associated with the continuation of breastfeeding under 24 months of age?

The hypothesis is that mothers who receive lactation counseling are less likely to engage in PLF practices and more likely to continue breastfeeding up to two years or beyond. Additionally, mothers experiencing food insecurity are more likely to adopt bottle feeding at an earlier stage.

This study aimed to investigate bottle feeding practices, food insecurity, the food diversity provided by Syrian mothers to their children under the age of two, and lactation counselling given to mothers, as well as the factors associated with PLF and continued breastfeeding. The results thus obtained are expected to shed light on maternal feeding practices, providing valuable insights for the enhancement of complementary feeding strategies for infants of Syrian refugee mothers in Türkiye.

## 2. Materials and Methods

### 2.1. Study Design

This cross-sectional descriptive study was carried out in collaboration with the Ministry of Health of Türkiye and Hacettepe University between 1 September 2020 and 30 December 2020. This study was conducted according to the guidelines laid down in the Declaration of Helsinki and all procedures involving research study participants were approved by the Ethical Board of Hacettepe University (decision number 2019/14-33). Written informed consent was obtained from mothers. If the mother was under 18 years, an adult caregiver gave permission.

### 2.2. Study Population

Syrian mothers who had children under the age of two and applied to selected refugee health centers (RHCs) on the survey day were included in the study. Mothers with more than one child under the age of two were excluded from the study because they confused the nutritional characteristics of the two children. Children having a history of prematurity (gestational duration < 37 weeks), low birth weight (<2500 g), hospitalization, and health problems affecting food consumption (e.g., phenylketonuria, spinal muscular atrophy) were not included in the study.

### 2.3. Sample Size

Considering the population size (*N*) of 500,000, the hypothesized percentage frequency of the outcome factor in the population (*p*): 50%, the confidence limits pf (*d*): 7.5%, a design effect (*DEFF*) of 1.2, with an equation: $$n=\frac{DEFF\times Np(1-p)}{d^{2}/z_{1-\alpha /2}^{2}\times \left(n-1\right)+p(1-p)}$$
the sample size for frequency in a population was calculated as 205 within the 95% confidence level [15].

### 2.4. Sampling

A three-stage sampling technique was used to select the Syrian refugee mothers; province, RHC, and mothers. One province was selected from the five Nomenclature of Territorial Units for Statistics (NUTS) regions in Türkiye, where there is a dense refugee settlement. Then the RHCs with the highest number of immigrant applications for health services in the provinces were selected (provinces (number of RHCs): İstanbul (12), Hatay (12), İzmir (6), Gaziantep (6), and Ankara (3)). A fieldwork day was set for each RHC. The first 6 mothers who applied on that day to the and volunteered to fill out the questionnaire were included in the study. Refugee health centers were considered as the sampling unit. In total, 39 different refugee health centers (RHCs) from 5 different provinces were selected through the Turkish Ministry of Health. In total, 234 survey forms were collected. After the mothers with two or three children under two years of age were excluded, 210 mother–infant dyads’ data were evaluated.

A survey form was created, questioning the feeding characteristics of Syrian infants. A Syrian health worker from each RHC administered the survey file to the mothers via face-to-face interviews. Mothers under 18 years were excluded if they did not have an adult caregiver or guardian.

### 2.5. Data Collection

A survey form was created based on the findings of previous studies [3,5]. It was translated into Arabic by bilingual (Turkish–Arabic) translators. Afterward, the survey was translated back into Turkish and checked. The survey consisted of 34 questions. In the first part, the demographic characteristics of the mother–infant dyad were questioned, and in the second part, breastfeeding and infant feeding characteristics were asked about. The first section collected demographic data on the mother–infant dyad, including the child’s age, gender, birth order, parental age, maternal education, and household composition. The second section focused on breastfeeding and infant feeding practices, addressing aspects such as lactation counseling during pregnancy and postpartum, the presence and composition of PLF, bottle feeding, pacifier usage, and food intake patterns of children. Lactation counselling is defined as guidance provided by a healthcare professional for the mother of the most recent child.

To evaluate the dietary diversity, all foods from different food groups (e.g., grains, fruits, vegetables, proteins, dairy, etc.) were assessed and coded. For example, we asked, “Which of the following vegetables has your child eaten or tasted before?”. According to the mother’s answers, the vegetables that the child ate were marked. Afterward, the question “Has your child eaten any vegetables other than the ones we have mentioned before?” was asked so as not to skip a vegetable that was given locally. The answers were noted. The translation of the forms was conducted by two Syrian healthcare personnel.

### 2.6. Statistical Analysis

The data were analyzed using the IBM-SPSS 23.0 program (SPSS Inc., Chicago, IL, USA). Descriptive statistics were used to present continuous variables as the mean ± standard deviation and quartiles (median (q1–q3)). Categorical variables were given as the frequency and percentage. The Chi-square test was used for comparisons between categorical variables. Multivariate logistic regression analysis determined the factors associated with PLF and continuing breastfeeding after controlling confounding factors. Variables with *p*-values < 0.10 in the Chi-square test were included in the models as confounding factors. A significance level of *p* < 0.05 was used.

## 3. Results

The mean age of the children was 11.0 ± 6.2 months and 45.2% were female; 22.9% of children were first-born. Mean maternal age was 27.1 ± 6.3 years; mean paternal age was 32.3 ± 6.9 years. The mothers were married at a mean age of 19.8 ± 4.1 years. While 8.1% of the mothers were illiterate, 14.8% were university graduates. Only 11.9% of the mothers were working and 7.6% were also pregnant during the study period. The mean number of households was 5.7 ± 2.2. The mean number of children of the mothers was 2.7 ± 1.5.

Lactation counseling was given to 40.5% of the mothers in the prenatal period and to 49.5% in the postnatal period, and to 33.8% in both the prenatal and postnatal periods. Thirty percent of the children had a history of PLF. The most frequently used prelacteal foods were sugary water (42.9%), formula (39.7%), and anise (23.8%). While 47.1% of the babies were bottlefed, 36.2% were using pacifiers. Among infants under 24 months, 69% were still breastfed (Table 1).

Complementary feeding was started mostly with dairy products. Interestingly, 39.3% of 0–3-month-old babies and 53.3% of 4–5-month-old babies had consumed dairy products before. In children aged 16–23 months, this rose to 92.0%. This rate was 88.6% in infants aged 6–23 months. The least consumed food groups were vitamin A-rich fruits and vegetables, legumes and nuts, and flesh foods. Among 16–23-month-old children, 40% had never consumed vitamin A-rich fruits and vegetables before. In infants aged 6–23 months, only 52.1% of them had consumed this food group before. Only 77.8% of 0–23-month-old children had previously consumed at least five of the eight food groups. This rate varied by age group (Table 2).

When the factors related to PLF were examined, it was found that first birth was associated with high PLF rates (41.7% and 26.5%, respectively; *p* = 0.045). Having lactation counseling support only in the prenatal period did not make any difference in terms of PLF (23.5% vs. 34.4%, *p* = 0.092). On the other hand, the percentage of PLF was lower in children whose mothers had lactation counseling support in the postpartum period (21.2% vs. 38.7%, *p* = 0.006) or whose mothers had lactation counseling support in both the prenatal and postpartum periods (16.9% vs. 36.7%, *p* = 0.003). The PLF was 42.4% in bottle fed babies, and 18.9% in babies who did not use feeding bottle (*p* < 0.001). The PLF was detected as 25.5% in infants who continued to be breastfed, while this rate was 40.0% in those who were not breastfed (*p* = 0.034). Continued breastfeeding was positively associated with being under the age of 6 months (86.0% and 64.7%, *p* = 0.007), not using feeding bottles (80.2% and 56.6%, *p* < 0.001), and not having PLF (73.5% and 58.7%, *p* = 0.034) (Table 3).

According to the multivariate analysis, it was found that PLF was associated with bottle feeding (AOR: 2.61 (95% CI: 1.35–5.10)). In addition, not having lactation counseling support in the prenatal and postnatal periods increased the risk of PLF 2.79 times (95% CI: 1.33–5.84).

Being 0–5 months old increased the odds of continued breastfeeding by 3.85 times (95% CI: 1.47–10.10). Again, not using feeding bottles increased the odds of continued breastfeeding by 2.66 (95% CI: 1.35–5.21) (Table 4).

## 4. Discussion

According to this current study, 69% of 0–23-month-old children continued to be breastfed. The determinants of still being breastfed under 23 months were the infant’s age and the bottle feeding. Almost half of the infants were bottle feeding. This rate was similar in Turkish children [16]. Nearly one in three infants included in the study received prelacteal food. Sugar water was the most commonly used prelacteal food. The use of prelacteal foods such as sugar water and anise is indeed a cultural characteristic, and it is worth noting that the continuation of breastfeeding beyond 15 months significantly decreases, from three-quarters of mothers to one-third. A qualitative study has also highlighted the cultural factors influencing breastfeeding practices and PLF [3]. A study conducted in Lebanon on the nutritional characteristics of Syrian refugees, involving 114 mother–infant pairs, found a PLF rate of 62.5%. The most commonly used prelacteal food was formula (70%), followed by sugar water (31.4%) [7]. Both the PLF rate and formula use were higher in that study compared with ours. This difference may be attributed to variations in the socioeconomic characteristics of refugees in the two countries. Additionally, inadequate baby-friendly hospital practices or particularly violations of the international code on the marketing of breast milk substitutes, such as formula donations and free formula distribution, may also explain this disparity. According to a study in which Turkey Demographic and Health Survey (TDHS) data were analyzed, the rate of PLF in Turkish children was found to be 41%. Infant formula was preferred as the prelacteal food at a rate of 92%, and sugar water constituted 1% of the prelacteal foods [17]. Consuming only sugary water as a substitute for breastmilk increases carbohydrate intake without providing essential nutrients, depriving infants of the vital calcium, protein, and other nutrients found in breastmilk. This can lead to undernutrition, vitamin deficiencies, and, over time, tooth decay in babies [18]. Moreover, excessive and prolonged consumption of sugary drinks in place of a balanced diet may increase the risk of metabolic issues later in life, such as obesity, insulin resistance, hypertension, and metabolic syndrome [19]. In this study, having both prenatal and postnatal lactation counseling support was found to be a protective factor against PLF. PLF has also been associated with bottle feeding. Previous studies showed that the absence of lactation counseling support was associated with PLF [20,21] and bottle feeding [16,20].

Sustainable Development Goal 2 targets creating a world free of hunger by 2030 [22]. Food insecurity is widespread among refugees and is associated with mental and physical health problems [6,23,24]. According to the results of this study, 40% of 16–23-month-old children had not eaten fruit and vegetables rich in vitamin A before. This rate was 28% for legumes and nuts; 20% for flesh food; 14% for grains, roots, and tubers; and 10% for eggs. These results may indicate problems in providing household food security and minimal dietary diversity. In support of this, according to TDHS 2018 data, only 17% of 18–23-month-old non-breastfed Syrian babies ate legumes the day before the survey. This rate was 10% for flesh food, 44% for eggs, and 63% for dairy products [25]. These rates were much better in Turkish children [26]. In other countries hosting Syrian refugees, notable differences in food security are observed between the local populations and refugees. A study conducted with Syrian refugees in Lebanon found that only 17.9% of Syrian refugee children met the minimum dietary diversity requirements, compared with 30.9% of children from the Lebanese host community (*p* < 0.05) [27]. Similarly, a population-based study conducted across refugee settings in eastern and southern Africa and western and central African regions in 15 countries found a low prevalence of bottle feeding (3.8%) among infants aged 0–5 months, contrasting with the findings of our study. However, consistent with our results, the median prevalence of the timely introduction of solid and semisolid foods (6–8 m) and flesh food consumption (6–23 m) was low, at 51.8% and 16.1%, respectively [28]. The nutritional status of refugees may be influenced by the socioeconomic conditions and healthcare quality in the host countries, as well as the specific characteristics of refugee communities. The complex interplay of factors, such as cultural practices, access to healthcare, and socioeconomic conditions, underscores the need for targeted interventions to address these disparities.

This study has several limitations. Given that the study was conducted at RHCs, it primarily includes individuals who sought healthcare services. We recognize that the findings may differ in populations that do not actively seek healthcare, as factors such as awareness of nutrition, cultural practices, and food insecurity may vary. In addition, this study excluded unregistered refugees, who may experience even greater food insecurity, potentially introducing selection bias. Unregistered refugees also lack a fixed residence, which makes them difficult to reach. As a result, the findings may not be generalizable to all Syrian refugees in Türkiye. However, this study focused on provinces with high refugee populations, and employed a three-stage sampling approach to ensure representation from diverse areas. In 2021, there were 181 RHCs operating across 29 provinces in Türkiye, providing a wide range of services such as primary health care, immunization, pregnancy monitoring, child health assessments, micronutrient support, cancer screenings, reproductive health services, psychosocial support, and health education. The establishment of these centers has been supported by the European Union through the Facility for Refugees in Türkiye [29]. Future studies that include both healthcare-seeking and non-healthcare-seeking refugee populations could offer a more comprehensive understanding of the diversity of feeding practices and nutritional status across different segments of the refugee community.

This study did not use a Standardized Dietary Diversity Score (SDDS), which limits comparability with other research. The SDDS is particularly useful for evaluating dietary diversity in infants and young children, typically between the ages of 6 months and 2 years, a critical period for assessing nutritional adequacy and the introduction of complementary foods. However, the focus on a 24 h recall in this study would not have allowed us to assess the broader variety of food groups the children had been exposed to up to that point. Since many refugees tend to consume a limited number of food groups, relying solely on the past 24 h would not provide an insight into whether children under the age of two had been introduced to other food groups. Therefore, although this study does not use SDDS, the findings still indicate that a high percentage of the participants lived in food-insecure households and were unable to achieve lifelong optimum dietary diversity.

The study relied on a single survey, as it was not feasible to schedule follow-up appointments with the same mothers, which is a limitation. To maximize participant compliance and obtain the most accurate responses, the questionnaire was kept concise. Given that we asked about the dietary diversity of the child up until the day of the visit, a multi-day dietary recall was not necessary. While self-reported food intake is indeed prone to recall bias, it is important to note that recall bias is expected to be minimized for children under 2 years old, particularly during the period when they are being introduced to different foods. Furthermore, a multi-day recall would have extended the survey length considerably, which was not desirable in this context.

Another limitation of this study is the lack of anthropometric measurements, which prevented a direct assessment of the children’s nutritional status. However, it is important to note that a single set of anthropometric measurements in infants can only diagnose malnutrition or obesity at that particular moment, whereas growth failure or excessive growth can be detected through ongoing monitoring. The focus of this study was to assess nutritional adequacy in terms of food group diversity. To fully understand the clinical implications of any nutritional deficiencies identified, future studies should incorporate anthropometric measurements and examine developmental characteristics over time to correlate feeding practices with actual health outcomes.

This study did not account for potential confounding factors such as socioeconomic status and employment. However, all participants had access to free healthcare services at RHCs, and many are recipients of assistance from the European Union, which may reduce the variability in healthcare access. Additionally, socioeconomic and employment-related data may not be accurate, as the participants, primarily mothers, may not have provided reliable responses regarding their financial situation or employment status. Future studies could address this limitation by incorporating more precise data on socioeconomic factors, though it remains challenging to obtain fully reliable information with a sensitivity analysis in this context.

The strength of this study lies in its comprehensive evaluation of infant nutrition, encompassing both breastfeeding and complementary foods. It effectively highlights the imbalanced food variety among Syrian refugee infants. However, it is important to acknowledge certain limitations. Firstly, this research is based on a cross-sectional design, limiting our ability to establish causation or examine changes over time. Additionally, reliance on maternal self-reporting introduces the potential for recall bias, as it lacks direct observations or follow-up data. Furthermore, language barriers were overcome with the assistance of interpreters, but using a translator could introduce interpretation nuances. Despite these limitations, this study provides valuable insights into the nutritional landscape of Syrian refugee infants in Türkiye.

## 5. Conclusions

Sugar water intake, PLF, bottle feeding, and insufficient MDD in complementary foods emerged as key concerns for Syrian infants. If mothers replace proper feeding, i.e., breastfeeding, with sugar water, it may disrupt glucose homeostasis. Additionally, it may reduce the baby’s desire to suckle at the breast, potentially leading to micro- and macronutrient deficiencies and malnutrition. Providing culturally tailored counseling is crucial in addressing this issue. Encouragingly, three out of four Syrian refugee infants exhibited satisfactory dietary variety, with nearly half of mothers incorporating legumes, nuts, and vitamin A-rich fruits and vegetables into their children’s diets. Notably, a quarter of Syrian babies had unrestricted access to formula milk, signaling a need for stricter control over formula milk support for refugees in Türkiye. The possibility of the impact of formula intake on dietary diversity raises considerations for interventions. Addressing this, the study underscores the pivotal role of lactation counseling in promoting proper breastfeeding practices and preventing PLF. Moving forward, there is a recognized opportunity to enhance the impact of lactation counseling by expanding its focus to include the principles of complementary feeding. By redefining it as “Infant and Young Child Feeding”, such training can empower mothers utilizing RHCs to offer a nutritionally diverse range of foods, steering them away from formula milk. Beyond lactation counseling, it is crucial to extend culturally tailored education to families, emphasizing the principles of complementary feeding and fostering a foundation for healthy child nutrition. This comprehensive approach is essential to ensure the holistic well-being of Syrian refugee infants and their families in Türkiye.

## Figures and Tables

**Table 1 nutrients-17-00684-t001:** Characteristics of mother–child pairs.

	Mean + SD (Range)/ N (%)	Median (q1–q3)
Child’s age (months)	11.0 ± 6.2 (0.3–23)	10.0 (6.0–15.0)
Gender, female	95 (45.2)	
Birth order, first	48 (22.9)	
Maternal age (years)	27.1 ± 6.3 (15–45)	26.0 (22.0–32.0)
Paternal age (years)	32.3 ± 6.9 (18–60)	31.0 (27.0–36.0)
Mothers’ age at marriage	19.8 ± 4.1 (11–39)	19.0 (17.0–21.0)
Maternal education		
Illiterate	17 (8.1)	
Literate	12 (5.7)	
Primary school	47 (22.4)	
Middle school	62 (29.5)	
High school	40 (19)	
University	31 (14.8)	
Working mothers	25 (11.9)	
Pregnant mothers	16 (7.6)	
Number of households	5.7 ± 2.2 (3–13)	5.0 (4.0–7.0)
Number of children	2.7 ± 1.5 (1–8)	2.0 (2.0–4.0)
Lactation counseling in pregnancy	85 (40.5)	
Lactation counseling in postpartum period	104 (49.5)	
Lactation counseling in both periods	71 (33.8)	
Prelacteal feeding	63 (30.0)	
• Sugar water	27 (42.9) *	
• Formula	25 (39.7) *	
• Anise	15 (23.8) *	
• Water	8 (12.7) *	
• Ayran	3 (4.8) *	
• Dates	2 (3.2) *	
• Cumin tea	2 (3.2) *	
Bottle feeding	99 (47.1)	
Pacifier usage	76 (36.2)	
Still breastfed	145 (69.0)	

* In those having prelacteal food.

**Table 2 nutrients-17-00684-t002:** Food intake of children by age.

	Age, Months							
	0–3	4–5	6–8	9–11	12–15	16–23	0–23	6–23
*n*	28	15	43	39	44	50	210	167
Breast milk, % *	85.7	86.7	76.5	79.5	75.0	36.0	69.0	64.7
Dairy products, %	39.3	53.3	88.2	87.2	86.4	92.0		88.6
Eggs, %	0.0	6.7	55.9	74.4	81.8	90.0		77.2
Flesh foods, %	0.0	0.0	41.2	61.5	77.3	80.0		67.1
Grains, roots, and tubers, %	0.0	0.0	79.4	76.9	88.6	86.0		83.2
Legumes and nuts, %	0.0	0.0	20.6	48.7	61.4	72.0		53.3
Vitamin A-rich fruits and vegetables, %	0.0	0.0	48.0	48.7	47.7	60.0		52.1
Other fruits and vegetables, %	3.6	20.0	97.1	87.2	97.7	96.0		94.6
Consuming ≥5 of 8 groups, %			58.8	76.9	86.4	84.0		77.8
Formula intake, %	32.9	13.3	29.4	33.3	18.2	16.0	23.8	23.4

* Breastmilk shows the proportion of children who continued to be breastfed. Other parameters show the percentages of eating or tasting that food before.

**Table 3 nutrients-17-00684-t003:** Factors associated with prelacteal feeding and continued breastfeeding, n = 210.

		Prelacteal Feeding		Still Breastfed	
	Overall, *n* (%) *	% **	*p* Value	% **	*p* Value
Child’s age (months)					
0–5	43 (20.5)	39.5	0.126	86.0	**0.007**
≥6	167 (79.5)	27.5		64.7	
Gender					
Female	95 (45.2)	30.5	0.880	64.2	0.168
Male	115 (54.8)	29.6		73.0	
Birth order					
First child	48 (22.9)	41.7	**0.045**	60.4	0.141
≥2 children	162 (77.1)	26.5		71.6	
Current maternal age (year)					
<25	88 (41.9)	37.5	0.116	72.7	0.587
25–30	62 (29.5)	22.6		67.7	
>30	60 (28.6)	26.7		65.0	
Maternal age at marriage (year)					
<19	93 (44.3)	32.3	0.524	72.0	0.403
≥19	117 (55.7)	30.0		66.7	
Maternal education					
Primary school and below	76 (36.4)	28.9	0.660	72.4	0.593
Middle school	62 (29.7)	33.9		64.5	
High school and above	71 (34.0)	26.8		70.4	
Lactation counseling in pregnancy					
Yes	85 (40.5)	23.5	0.092	71.8	0.482
No	125 (59.5)	34.4		67.2	
Lactation counseling in postpartum period					
Yes	104 (49.5)	21.2	**0.006**	73.1	0.211
No	106 (50.5)	38.7		65.1	
Lactation counseling in both pregnancy and postpartum periods					
Yes	71 (33.8)	16.9	**0.003**	73.2	0.348
No	139 (66.2)	36.7		66.9	
Bottle feeding					
Yes	99 (47.1)	42.4	**<0.001**	56.6	**<0.001**
No	11 (52.9)	18.9		80.2	
Pacifier usage					
Yes	76 (36.2)	34.2	0.316	61.8	**0.089**
No	134 (63.8)	27.6		73.1	
Prelacteal feeding					
Yes	63 (30.0)			58.7	**0.034**
No	147 (70.0)			73.5	
Still breastfed					
Yes	145 (69.0)	25.5	**0.034**		
No	65 (31.0)	40.0			

* Row percentage; ** Column percentage.

**Table 4 nutrients-17-00684-t004:** Factors associated with prelacteal feeding and continued breastfeeding (multiple logistic regression, *n* = 210).

Determinants of Prelacteal Feeding	AOR (95% CI) *
Birth order	
First	1.78 (0.86–3.67)
≥2nd child	1
Lactation counseling in both periods ^#^	
Yes	1
No	**2.79 (1.33–5.84)**
Bottle feeding	
Yes	**2.61 (1.35–5.10)**
No	1
Still breastfed	
Yes	0.72 (0.37–1.43)
No	
**Determinants of being still breastfed**	**AOR (95% CI) ****
Age, months	
0–5	**3.85 (1.47–10.10)**
≥6	1
Bottle feeding	
Yes	1
No	**2.66 (1.35–5.21)**
Pacifier usage	
Yes	1
No	1.19 (0.61–2.31)
Prelacteal feeding	
Yes	0.57 (0.29–1.12)
No	1

^#^ Pregnancy and postpartum period. * Birth order, breastfeeding counseling in both pregnancy and the postpartum period, bottle feeding, and being still breastfed were included in the model. ** Bottle feeding, age, pacifier usage, and prelacteal feeding were included in the model.

## Data Availability

The datasets are available from the corresponding author on request. The data are not publicly available due to ethical reasons.

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
