# Peer review of "Unraveling Infant Feeding Practices Among Syrian Refugees in Türkiye: A Cross-Sectional Study"

_nutrients, 2025, doi:10.3390/nu17040684_

Round 1

Reviewer 1 Report

Comments and Suggestions for Authors

Overall Assessment

The study explores a crucial and under-researched topic: infant feeding practices among Syrian refugees in Türkiye. It provides valuable insights into breastfeeding trends, dietary diversity, and challenges faced by refugee mothers. The research is well-structured, utilizes sound methodology, and is grounded in relevant literature.

However, there are certain areas that require improvements for clarity, rigor, and impact. Below are the strengths and weaknesses, along with recommendations for enhancement.

Positive Aspects

  1. Relevance and Contribution to Literature
    • The topic is highly relevant, given the growing number of Syrian refugees and the vulnerability of infants to malnutrition.
    • It builds upon existing studies but provides new insights into complementary feeding practices, dietary diversity, and socio-cultural influences.
  2. Methodological Strengths
    • The study employs a three-stage sampling technique, ensuring diversity in the sample.
    • The use of multivariate logistic regression strengthens the analysis of factors influencing feeding practices.
    • The inclusion of Syrian healthcare workers for data collection improves reliability by minimizing language and cultural barriers.
  3. Clear Presentation of Results
    • Data is well-organized in tables and statistical tests are appropriately applied.
    • The findings are statistically sound, with p-values and confidence intervals provided.
    • Associations between prelacteal feeding, breastfeeding, and dietary diversity are clearly explained.
  4. Public Health Implications
    • The study highlights key concerns such as high rates of prelacteal feeding, bottle-feeding, and lack of dietary diversity.
    • The emphasis on lactation counseling and culturally appropriate nutritional education is a strong policy recommendation.

Areas for Improvement

  1. Clarity in Research Objectives
    • The introduction should more explicitly define the research questions and hypotheses.
    • What are the specific gaps in literature that this study aims to fill? Providing a clearer statement would enhance focus.
  2. Study Limitations & Generalizability
    • The authors mention that the study cannot be generalized to all Syrian refugees. However, additional discussion on how representative the sample is and how results might vary in non-healthcare-seeking populations would strengthen validity.
    • The study excludes unregistered refugees, who might face even greater food insecurity. This should be acknowledged as a potential bias.
  3. Dietary Diversity Analysis
    • The study does not use a standardized dietary diversity score (DDS), which limits comparability with other research.
    • Self-reported food intake is prone to recall bias. The authors acknowledge this but should discuss alternative approaches, such as multiple-day dietary recall.
  4. Anthropometric Data
    • The study does not include anthropometric measurements (weight, height, growth charts). This is a major limitation since it prevents assessing the direct nutritional status of the children.
    • Future research should incorporate malnutrition indicators (e.g., stunting, wasting, underweight) to correlate feeding practices with actual health outcomes.
  5. Statistical Considerations
    • The study does not account for potential confounding factors such as socioeconomic status, employment, and access to healthcare.
    • A sensitivity analysis could help assess whether results remain stable under different assumptions.
  6. Discussion & Comparison with Existing Literature
    • While the discussion is strong, a deeper comparison with global studies on refugee infant nutrition (e.g., studies from Lebanon, Jordan, and European resettlement programs) would enhance contextual understanding.
    • The study could explore the role of cultural beliefs in greater detail, as these often shape feeding practices.
Comments on the Quality of English Language

The overall quality of English in the manuscript is acceptable but requires minor refinements for clarity, coherence, and academic precision. The writing is generally clear, well-structured, and grammatically sound, but some areas could benefit from improved phrasing, word choice, and sentence flow to enhance readability and professionalism.

Author Response

Response to Reviewer 1 Comments

Thank you very much for taking the time to review this manuscript. Please find the detailed responses below and the corresponding revisions highlighted changes in the re-submitted files.

Comments 1: The introduction should more explicitly define the research questions and hypotheses.

Response 1: Thank you for your comment. “This study aims to answer the following research questions:

*What are the bottle-feeding practices among Syrian refugee mothers with children under the age of two?

*What is the extent of food insecurity in infant feeding practices experienced by Syrian refugee families?

*What is the level of food diversity provided by Syrian mothers to their children under the age of two?

*What is the prevalence of lactation counseling received by Syrian refugee mothers?

*What factors are associated with PLF among Syrian refugee children?

*What factors are associated with the continuation of breastfeeding under 24 months of age?

The hypothesis is that mothers who receive lactation counseling are less likely to engage in PLF practices and more likely to continue breastfeeding up to two years or beyond. Additionally, mothers experiencing food insecurity are more likely to adopt bottle-feeding at an earlier stage.” Was added to introduction (Line 68-82).

Comments 2: What are the specific gaps in literature that this study aims to fill? Providing a clearer statement would enhance focus.

Response 2: Thank you for your comment. We added to the introduction section lines 59-67 “There are some studies investigating the breastfeeding and infant feeding characteris-tics of Syrian refugee children in Türkiye [2-5]. Additionally, food insecurity is a prev-alent challenge among refugee populations, yet studies investigating the food security status of refugees in Türkiye remain limited [13,14]. Despite these efforts, there is a significant gap in the literature, as no comprehensive study has holistically examined breastfeeding practices, prelacteal feeding (PLF), and complementary food diversity among Syrian refugee children. Addressing this gap is essential for developing target-ed strategies to improve infant nutrition and maternal feeding practices within this vulnerable population.”

Comments 3: The authors mention that the study cannot be generalized to all Syrian refugees. However, additional discussion on how representative the sample is and how results might vary in non-healthcare-seeking populations would strengthen validity.

Response 3: Thank you for your suggestion. While we acknowledge that the study cannot be generalized to all Syrian refugees, we have made efforts to ensure that the sample is representative of the refugee population residing in provinces with dense refugee settlements in Turkey. Given that the study was conducted at Refugee Health Centers (RHC), it primarily includes individuals who seek healthcare services. We recognize that the findings may differ in populations that do not actively seek healthcare, as factors such as awareness of nutrition, cultural practices, and food insecurity may vary. Future studies that include both healthcare-seeking and non-healthcare-seeking refugee populations could offer a more comprehensive understanding of the diversity of feeding practices and nutritional status across different segments of the refugee community.

Comments 4: The study excludes unregistered refugees, who might face even greater food insecurity. This should be acknowledged as a potential bias.

Response 4: Thank you for your suggestion, we agree with this comment . Therefore, we have added to discussion section limitation part, lines 271-282 “This study excluded unregistered refugees, who may experience even greater food in-security, potentially introducing selection bias. Unregistered refugees also lack a fixed residence, which makes them difficult to reach. As a result, the findings may not be generalizable to all Syrian refugees in Türkiye. However, the study focused on provinces with high refugee populations, and a three-stage sampling approach was used in Refugee Health Centers (RHCs) to ensure representation from diverse areas. In 2021, there were 181 RHCs operating across 29 provinces in Türkiye, providing a wide range of services such as primary health care, immunization, pregnancy monitoring, child health assessments, micronutrient support, cancer screenings, reproductive health ser-vices, psychosocial support, and health education. The establishment of these centers has been supported by the European Union through the Facility for Refugees in Türkiye[29].  

Comments 5: The study does not use a standardized dietary diversity score (DDS), which limits comparability with other research.

Response 5: We have added to discussion section limitation part, lines 286-296, “The study did not use a Standardized Dietary Diversity Score (SDDS), which lim-its comparability with other research. The SDDS is particularly useful for evaluating dietary diversity in infants and young children, typically between the ages of 6 months to 2 years, a critical period for assessing nutritional adequacy and the introduction of complementary foods. However, the focus on a 24-hour recall in this study would not have allowed us to assess the broader variety of food groups the children had been ex-posed to up to that point. Since many refugees tend to consume a limited number of food groups, relying solely on the past 24 hours would not provide insight into whether children under the age of two had been introduced to other food groups. Therefore, although the study does not use SDDS, the findings still indicate that a high percentage of the participants lived in food-insecure households and were unable to achieve lifelong optimum dietary diversity.”

Comments 6: Self-reported food intake is prone to recall bias. The authors acknowledge this but should discuss alternative approaches, such as multiple-day dietary recall.

Response 6: “The study relied on a single survey, as it was not feasible to schedule follow-up appointments with the same mothers, which is a limitation. To maximize participant compliance and obtain the most accurate responses, the questionnaire was kept con-cise. Given that we asked about the dietary diversity of the child up until the day of the visit, a multi-day dietary recall was not necessary. While self-reported food intake is indeed prone to recall bias, it is important to note that recall bias is expected to be minimized for children under 2 years old, particularly during the period when they are being introduced to different foods. Furthermore, a multi-day recall would have ex-tended the survey length considerably, which was not desirable in this context..” was added to the limitation section lines 297-305.

Comments 7: The study does not include anthropometric measurements (weight, height, growth charts). This is a major limitation since it prevents assessing the direct nutritional status of the children. Future research should incorporate malnutrition indicators (e.g., stunting, wasting, underweight) to correlate feeding practices with actual health outcomes

Response 7: This part was revised as “Another limitation of the study is the lack of anthropometric measurements, which prevents a direct assessment of the children's nutritional status. However, it is important to note that a single set of anthropometric measurements in infants can only diagnose malnutrition or obesity at that particular moment, whereas growth failure or excessive growth can be detected through ongoing monitoring. The focus of this study was to assess nutritional adequacy in terms of food group diversity. To fully under-stand the clinical implications of any nutritional deficiencies identified, future studies should incorporate anthropometric measurements and examine developmental char-acteristics over time to correlate feeding practices with actual health outcomes..” in lines 306-314

Comments 8: The study does not account for potential confounding factors such as socioeconomic status, employment, and access to healthcare. A sensitivity analysis could help assess whether results remain stable under different assumptions.

Response 8: Thank you for your suggestion, “This study did not account for potential confounding factors such as socioeco-nomic status, and employment. However, all participants had access to free healthcare services at RHCs, and many are recipients of assistance from the European Union, which may reduce the variability of healthcare access. Additionally, socioeconomic and employment-related data may not be accurate, as participants, primarily mothers, may not have provided reliable responses regarding their financial situation or em-ployment status. Future studies could address this limitation by incorporating more precise data on socioeconomic factors with sensitivity analysis, though it remains challenging to obtain fully reliable information in this context.” Added to lines 315-322.

Comments 9: While the discussion is strong, a deeper comparison with global studies on refugee infant nutrition (e.g., studies from Lebanon, Jordan, and European resettlement programs) would enhance contextual understanding.

Response 9: Thank you for your comment we added to lines 252-266 “In other countries hosting Syrian refugees, notable differences in food security are ob-served between local populations and refugees. A study conducted with Syrian refu-gees in Lebanon found that only 17.9% of Syrian refugee children met minimum die-tary diversity requirements, compared to 30.9% of children from the Lebanese host community (p < 0.05) [27]. Similarly, a population-based study conducted across refugee settings in Eastern and Southern Africa and Western and Central Africa regions in 15 countries found a low prevalence of bottle-feeding (3.8%) among infants aged 0–5 months, contrasting with the findings of our study. However, consistent with our results, the median prevalence of timely introduction of solid and semisolid foods (6-8 m) and flesh food consumption (6-23 m) was low, at 51.8% and 16.1%, respectively [28]. The nutritional status of refugees may be influenced by the socioeconomic conditions and healthcare quality in host countries, as well as the specific characteristics of refugee communities. The complex interplay of factors, such as cultural practices, access to healthcare, and socioeconomic conditions, underscores the need for targeted interventions to address these disparities.”

Comments 10: The study could explore the role of cultural beliefs in greater detail, as these often shape feeding practices.

Response 10: Thank you for your comment. “The use of prelacteal food as sugar water and anise is indeed a cultural characteristic, and it is worth noting that the continuation of breastfeeding beyond 15 months significantly decreases, from three-quarters of mothers to one-third. A qualitative study have also highlighted the cultural factors influencing breastfeeding practices and prelacteal feeding [3]. A study conducted in Lebanon on the nutritional characteristics of Syrian refugees, involving 114 mother-infant pairs, found a prelacteal feeding rate of 62.5%. The most commonly used prelacteal food was formula (70%), followed by sugar water (31.4%)[7]. Both the prelacteal feeding rate and formula use were higher in this study compared to ours. This difference may be attributed to variations in the socioeconomic characteristics of refugees in the two countries. Additionally, baby-friendly hospital practices, particularly violations of the international code on the marketing of breast milk substitutes, such as formula donations and free formula distribution, may also discussed in this disparity. ……… Consuming only sugary water as a substitute for breastmilk increases carbohydrate intake without providing essential nutrients, depriving of vital calcium, protein, and other nutrients found in breastmilk. This can lead to undernutrition, vitamin deficiencies, and, over time, tooth decay in babies[18]. Moreover, excessive and prolonged consumption of sugary drinks in place of a balanced diet may increase the risk of metabolic issues later in life, such as obesity, insulin resistance, hypertension, and metabolic syndrome[19].” Was added to lines 218-238.

Comments on the Quality of English Language: The overall quality of English in the manuscript is acceptable but requires minor refinements for clarity, coherence, and academic precision. The writing is generally clear, well-structured, and grammatically sound, but some areas could benefit from improved phrasing, word choice, and sentence flow to enhance readability and professionalism.

Response: Thank you for your feedback. The overall quality of English in the manuscript has been reviewed and refined for clarity, coherence, and academic precision.

Reviewer 2 Report

Comments and Suggestions for Authors

This research could have a great potential, but it wasn't used. Limitations of the study provided by authors explain all: many questions which should be asked, but they weren't; measurements which should be done, and again, they weren't. With limited amount of data gathered during the research, I think authors did what they could to present the feeding of babies and infants of Syrian refugees. The only section which can be significantly improved to increase the value of this manuscript is discussion. In this form discussion is short and very plain. All paragraphs should be extended (there are only 3 of them directly concerning results) with proper comparison to other published research including PLF and food groups, as both issues were treated superficially. Conclusion section: The problem of formulated milk in PLF, contrary to the authors' statement, is not the biggest issue in newborns feeding. I my opinion bigger problem here, is feeding with sugary water:  it only increases carbohydrates load  but gives no additional value to the meal. If used often as replacement for proper feeding, it may cause the problems with glucose homeostasis in later life and also  may cause malnutrition and developmental problems. And that should be clearly stated.

Author Response

We are grateful for your constructive input in guiding, for your valuable suggestions and the time you dedicated to reviewing our work.

Comments 1: This research could have a great potential, but it wasn't used. Limitations of the study provided by authors explain all: many questions which should be asked, but they weren't; measurements which should be done, and again, they weren't. With limited amount of data gathered during the research, I think authors did what they could to present the feeding of babies and infants of Syrian refugees. The only section which can be significantly improved to increase the value of this manuscript is discussion. In this form discussion is short and very plain. All paragraphs should be extended (there are only 3 of them directly concerning results) with proper comparison to other published research including PLF and food groups, as both issues were treated superficially.

Response 1: Thank you for pointing this out.

We added lines 189-197 “In a study examining the nutritional characteristics of Syrian refugees in Lebanon, which included 114 mother-infant pairs, the rate of prelacteal feeding was 62.5%. However, the most commonly used prelacteal food was formula (70%), followed by sugar water (31.4%)[7]. In the study conducted in Lebanon, both the prelacteal feeding rate and formula use rate were higher compared to our study. This difference may be due to the difference in socioeconomic characteristics of refugees in both countries. In addition, baby-friendly hospital practices in both countries, especially violations of the international code of marketing of breast milk substitutes such as formula donations and free formula distribution, may also be an important reason for this difference.”

An line 220-29 “In other countries hosting Syrian refugees, notable differences in food security are ob-served between local populations and refugees. A study conducted with Syrian refu-gees in Lebanon found that only 17.9% of Syrian refugee children met minimum die-tary diversity requirements, compared to 30.9% of children from the Lebanese host community (p < 0.05) [27]. Similarly, a population-based study conducted across refu-gee settings in Eastern and Southern Africa and Western and Central Africa regions in 15 countries found a low prevalence of bottle-feeding (3.8%) among infants aged 0–5 months, contrasting with the findings of our study. However, consistent with our re-sults, the median prevalence of timely introduction of solid and semisolid foods (6-8 m) and flesh food consumption (6-23 m) was low, at 51.8% and 16.1%, respectively [28]. The nutritional status of refugees may be influenced by the socioeconomic conditions and healthcare quality in host countries, as well as the specific characteristics of refugee communities. The complex interplay of factors, such as cultural practices, ac-cess to healthcare, and socioeconomic conditions, underscores the need for targeted interventions to address these disparities.”

Comments 2: Conclusion section: The problem of formulated milk in PLF, contrary to the authors' statement, is not the biggest issue in newborns feeding. I my opinion bigger problem here, is feeding with sugary water:  it only increases carbohydrates load  but gives no additional value to the meal. If used often as replacement for proper feeding, it may cause the problems with glucose homeostasis in later life and also  may cause malnutrition and developmental problems. And that should be clearly stated.

Response 2: We agree with this comment. Therefore, we added discussion part lines 101-107 “Consuming only sugary water as a substitute for breastmilk increases carbohydrate intake without providing essential  nutrients, depriving of vital calcium, protein, and other nutrients found in breastmilk. This can lead to undernutrition, vitamin deficiencies, and, over time, tooth decay in babies [18]. Moreover, excessive and prolonged consumption of sugary drinks in place of a balanced diet may increase the risk of metabolic issues later in life, such as obesity, insulin resistance, hypertension, and metabolic syndrome [19].” And conclusion section lines 271-274 “If mothers replace proper feeding, breastfeeding, with sugar water, it may disrupt glucose homeostasis. Additionally, it may reduce the baby's desire to suckle at the breast, potentially leading to micro and macronutrient deficiencies, malnutrition. Providing culturally tailored counseling is crucial in addressing this issue..”

Reviewer 3 Report

Comments and Suggestions for Authors

I have read this paper with great interest, and confirm that the topic has its relevance. I only have specific additional reflections for additional specific revisions. 

Representativity of the cohort ? The inclusion criteria are clear (applied to selected RHCs on the survey day, but are all Syrian refugees serves by these services, or is this given subgroup. This could be relevant to understand the data as good as possible, and can be addressed by e.g. be clearer on these limitations in the title, abstract and full paper ?

We also need more information on the survey form. Based on the introduction of the text on WHO, the survey is very likely informed by these WHO documents, but this is not yet sufficiently clear.

Is here any information on who provided the lactation counselling (as it seems that this is a cohort with previous experience on breastfeeding, considering 22.9 % of the children were first borns).

Comments on the Quality of English Language

no specific comments on these aspects

Author Response

Thank you very much for taking the time to review this manuscript. Please find the detailed responses below and the corresponding revisions highlighted changes in the re-submitted files.

Comments 1: Representativity of the cohort? The inclusion criteria are clear (applied to selected RHCs on the survey day, but are all Syrian refugees serves by these services, or is this given subgroup. This could be relevant to understand the data as good as possible, and can be addressed by e.g. be clearer on these limitations in the title, abstract and full paper

Response 1: Thank you for your insightful comment.

The inclusion criteria were indeed applied to a selected group of Refugee Health Centers (RHCs) on the survey day. These centers primarily serve Syrian refugees who actively seek healthcare services. As such, the study sample consists of healthcare-seeking individuals, which may not fully represent the broader Syrian refugee population in Turkey, particularly those who do not seek medical assistance. We acknowledge that this could influence the generalizability of our findings, particularly in relation to nutrition and feeding practices. To clarify these limitations, we will ensure that they are explicitly addressed in the title, abstract, and throughout the manuscript to provide a more accurate understanding of the study's scope and its limitations regarding representativity. We added “Given that the study was conducted at RHCs, it primarily includes individuals who seek healthcare services. We recognize that the findings may differ in populations that do not actively seek healthcare, as factors such as awareness of nutrition, cultural practices, and food insecurity may vary. In addition, this study excluded unregistered refugees, who may experience even greater food insecurity, potentially introducing se-lection bias. Unregistered refugees also lack a fixed residence, which makes them difficult to reach. As a result, the findings may not be generalizable to all Syrian refugees in Türkiye. However, the study focused on provinces with high refugee populations, and a three-stage sampling approach was used in Refugee Health Centers (RHCs) to ensure representation from diverse areas. In 2021, there were 181 RHCs operating across 29 provinces in Türkiye, providing a wide range of services such as primary health care, immunization, pregnancy monitoring, child health assessments, micronutrient support, cancer screenings, reproductive health services, psychosocial support, and health education. The establishment of these centers has been supported by the European Union through the Facility for Refugees in Türkiye [29].  Future studies that include both healthcare-seeking and non-healthcare-seeking refugee populations could offer a more comprehensive understanding of the diversity of feeding practices and nutritional status across different segments of the refugee community.” To limitation section.

Comments 2:  We also need more information on the survey form. Based on the introduction of the text on WHO, the survey is very likely informed by these WHO documents, but this is not yet sufficiently clear.

Response 2: Method section was revised; The research questions were created based on the findings of previous studies  [3,5]. The first section collected demographic data on the mother-infant dyad, including child age, gender, birth order, parental age, maternal education, and household composition. The second section focused on breastfeeding and infant feeding practices, ad-dressing aspects such as lactation counseling during pregnancy and postpartum, presence and composition of prelacteal feeding, bottle-feeding, pacifier usage, and food intake patterns of children.

Comments 3: Is here any information on who provided the lactation counselling (as it seems that this is a cohort with previous experience on breastfeeding, considering 22.9 % of the children were first borns).

Response 3: Lactation counselling is defined as guidance provided by a healthcare professional for the mother of the most recent child.

Round 2

Reviewer 2 Report

Comments and Suggestions for Authors

Authors answered all my notes. I have no further issues with this manuscript. Paper can be published.